# Analytical Validation of Cxbladder^®^ Detect, Triage, and Monitor: Assays for Detection and Management of Urothelial Carcinoma

**DOI:** 10.3390/diagnostics14182061

**Published:** 2024-09-17

**Authors:** Justin C. Harvey, Lisa M. Cambridge, Charles W. Ellen, Megan Colonval, Jody A. Hazlett, Jordan Newell, Xin Zhou, Parry J. Guilford

**Affiliations:** 1Pacific Edge Diagnostics NZ, Ltd., 87 St. David Street, Dunedin 9016, New Zealand; lisa.cambridge@pacificedgedx.com (L.M.C.); charles.ellen@pacificedge.co.nz (C.W.E.); megan.colonval@pacificedge.co.nz (M.C.); jody.hazlett@pacificedge.co.nz (J.A.H.); xin.zhou@pacificedge.co.nz (X.Z.); parry.guilford@otago.ac.nz (P.J.G.); 2Cambridge Quality Consulting, T/A Check Electric Ltd., 204 Coast Road, Waikouaiti 9471, New Zealand; 3Pacific Edge Diagnostics USA, Ltd., 1214 Research Boulevard, Hummelstown, PA 17036, USA; jordan.newell@pacificedgedx.com; 4Department of Biochemistry, University of Otago, 710 Cumberland Street, Dunedin 9016, New Zealand

**Keywords:** analytical validation, biomarkers, hematuria, monitoring, reverse transcription-quantitative PCR, urothelial carcinoma, urinary tests

## Abstract

**Background**: Cxbladder^®^ assays are reverse transcription-quantitative polymerase chain reaction (RT-qPCR) tests incorporating five genetic biomarkers (*CDK1*, *MDK*, *IGFBP5*, *HOXA13*, and *CXCR2*) to provide risk stratification for urothelial carcinoma (UC) in patients with hematuria or undergoing surveillance for recurrent disease. This study evaluated the analytical validity of the Cxbladder Detect, Triage, and Monitor assays. **Methods:** Pre-specified acceptance criteria, including the assays’ fundamental aspects (sample and reagent stability, RNA extraction quality, RT-qPCR linearity, and analytical sensitivity and specificity), accuracy and precision, and reproducibility between laboratories. **Results:** Cxbladder had an analytical sensitivity of 12.5–31.1 RNA copies/mL urine for the *CDK1*, *MDK*, *IGFBP5*, and *HOXA13* UC biomarkers and 68.9 RNA copies/mL for the inflammatory biomarker *CXCR2*. All the pre-specified analytical criteria were met. Cxbladder had diagnostic sensitivity, specificity, positive predictive value, and negative predictive values of 77%, 94%, 68%, and 96%, respectively, for Detect; 95%, 46%, 20%, and 98% for Triage; and 91%, 39%, 21%, and 96% for Monitor. Cxbladder had high analytical accuracy (≤10.63% inaccuracy across all biomarkers) and good reproducibility (>85% concordance between laboratories). **Conclusions:** Cxbladder accurately and reproducibly detects UC biomarker expression and can aid clinicians in risk stratification of hematuria patients or those undergoing surveillance for recurrent UC.

## 1. Introduction

Hematuria (or blood in the urine) is a common condition that often has a benign cause; however, it is one of the main symptoms of urothelial carcinoma (UC) [1]. Although the American Urological Association guidelines recommend that patients with hematuria undergo further clinical evaluation [2], the rates of confirmed UC in patients with hematuria are low (3.3% of those with asymptomatic microhematuria and ~17% of those with gross hematuria) [3]. Further, among patients with hematuria referred for further evaluation, ~18% may have atypical cytology or equivocal cystoscopy [4], thereby returning inconclusive results that require further invasive procedures (e.g., biopsy). In addition, as the recurrence rates for non-muscle-invasive bladder cancer are high, with 38–61% of patients having tumor recurrence after transurethral resection [5], ongoing guideline-directed surveillance is recommended in patients with a history of UC [6].

Cystoscopy is the gold standard for UC diagnosis [2,6], but both white- and blue-light cystoscopy give variable results and sensitivities of less than 100% (73–86% and 95–96%, respectively) [7], with false-positive rates of 12–45% and 15–37%, respectively [8]. Historically, the main non-invasive methods used for evaluating patients for UC (either initially or for recurrence) were urine cytology (which has high specificity) and fluorescence in situ hybridization; however, both of these methods have poor sensitivity [9,10]. This leads to high false-negative rates, meaning that neither method can safely rule out UC. Therefore, there is a need for accurate, non-invasive urinary biomarker tests that can rule out patients with hematuria who do not have UC and identify those with the highest risk of UC, including those at risk of recurrent tumors.

The Cxbladder^®^ assays are reverse transcription quantitative-PCR (RT-qPCR)-based diagnostic urinary biomarker tests that can be used for (a) risk stratification of patients presenting for evaluation of hematuria (Cxbladder Detect and Cxbladder Triage) and (b) the surveillance of patients with previously treated UC (Cxbladder Monitor) [11,12,13]. Cxbladder assays quantify the messenger RNA (mRNA) expression of five biomarkers, including four genes that are known to be correlated with UC (cyclin-dependent kinase 1 [*CDK1*], midkine [*MDK*], insulin-like growth factor binding protein 5 [*IGFBP5*], and Homeobox A13 [*HOXA13*]) [14], plus a known marker of inflammation (C-X-C motif chemokine receptor 2 [*CXCR2*]) [11]. The *CXCR2* biomarker, which is highly expressed in neutrophils, is included in the Cxbladder assays to reduce the noise from background inflammation to help discriminate between individuals with nonmalignant disease and those with UC [11]. In previous studies in patients with microhematuria or gross hematuria, Cxbladder Detect demonstrated moderate sensitivity and high specificity to enable risk stratification for the presence or absence of UC [4,15,16], and Cxbladder Triage had high sensitivity and a high negative predictive value (NPV), providing a strong value proposition as a rule-out test (test-negative tool) [4,15,16,17]. These findings indicate that Cxbladder Detect and Triage may provide a more robust non-invasive method for detection of UC than urine cytology in patients with hematuria. In real-world clinical practice, the Cxbladder Detect assay can be used as the first step for evaluating patients with microhematuria to exclude those with negative results from further evaluation; urine cytology could then be conducted in patients with positive results to further improve the positive predictive value of the non-invasive evaluation for those patients. In patients undergoing surveillance for recurrent UC, Cxbladder Monitor also had high sensitivity and NPV [4], thereby allowing the physician to safely rule out recurrent UC in patients who may benefit from a reduced intensity of invasive procedures.

For biomarker tests to be incorporated into routine clinical use, they must demonstrate strong analytical validity, clinical validity, and clinical utility [18]. Prior to clinical validation, analytical validation studies of the Cxbladder assays were conducted to ensure that the analytical component of the assays were well controlled and provided reliable results with high precision and accuracy. The fundamental aspects of the assays (sample and reagent stability, RNA extraction quality, linearity of RT-qPCR, analytical sensitivity and specificity, and pre-analytical and analytical measurement of the five biomarker mRNA concentrations) are identical for all three assays and were assessed using Cxbladder Detect. The accuracy, precision, and reproducibility were then confirmed for each Cxbladder assay (Detect, Triage, and Monitor). Here, we report the results of this analytical validation study.

## 2. Materials and Methods

### 2.1. Sample Collection

For clinical use of Cxbladder assays, voided mid-stream urine samples (4.5 mL) are required. The analytical validation analysis used mid-stream urine samples collected from consenting healthy volunteers. For the clinical validation analysis of Cxbladder described here, voided mid-stream urine samples were collected from consenting patients presenting with hematuria (for Cxbladder Detect or Triage) or those undergoing surveillance for recurrence of UC (for Cxbladder Monitor). Clinical trial sample acquisition has been described in full previously for Cxbladder Detect [11], Triage [12], and Monitor [13].

All clinical trials were registered on ClinicalTrials.gov or the Australia New Zealand Clinical Trials Registry (ANZCTR) and adhered to the ethical principles of the Declaration of Helsinki of 2013. All sample collection was performed with written informed consent obtained from the participants of the study and was performed under local ethical approvals.

### 2.2. Sample and Reagent Stability

Urine samples were aspirated into a Cxbladder Urine Sampling System (USS) that contained Cxbladder stabilizing reagent (4.5 mL; a proprietary nucleic acid-stabilizing solution) and transported from the clinic to the Pacific Edge Diagnostics laboratories (Dunedin, New Zealand [NZ] or Hummelstown, PA, USA). To assess the stability of the sample and stabilizing reagent during transportation, 62 high- and low-extraction controls (HEC and LEC) with RNA-equivalent concentrations were added to Cxbladder stabilizing reagent, aliquoted, and stored at 2–8 °C, 18–22 °C (laboratory controlled ambient temperature), and 30 °C. To establish the baseline timepoint (T0), eight replicates were stored at each temperature for 2 h prior to being processed through the RNA extraction procedure described below. Aliquots stored at 2–8 °C, 18–22 °C, and 30 °C were assessed on Days 2, 5, 7, 9, 12, and 14 (in triplicate) and compared against the T0 samples.

The freeze/thaw stability of the Cxbladder stabilizing reagent was assessed by comparing two aliquots of reagents: aliquots that were frozen (at −80 °C) and thawed five times versus non-frozen aliquots.

The freeze/thaw stability of the RNA was assessed using reference RNA. The reference RNA was a plasmid-derived RNA transcript comprising the five biomarker targets measured in all Cxbladder assays (described in detail below). Four aliquots of RNA were stored at −80 °C (optimal condition) and an additional four aliquots underwent three −80 °C freeze/thaw cycles (a >12-h freeze step and a 1-h thaw step). Each reference RNA aliquot was then diluted to create a four-point standard curve in an RT-qPCR plate, with each serially diluted point plated in triplicate. Assessment of the sample stability of the extractions was performed using previously published methods [14].

### 2.3. RNA Extraction Quality

Analytical validation of RNA extractions was conducted using an automated, silica matrix-coated, magnetic bead-based system. Briefly, 2.9 mL aliquots of stabilized sample were added to 1.1 mL of absolute ethanol containing 25 µL of MagMAX beads (Thermo Fisher Scientific Inc., Waltham, MA, USA), 3.625 µg of poly-A carrier RNA (Roche, Basel, Switzerland) and 17,260 copies of an exogenous control RNA (Cells-to-CT^TM^ Control Kit; Thermo Fisher Scientific Inc., Waltham, MA, USA). Samples were extracted on a KingFisher^TM^ Apex instrument (Thermo Fisher Scientific Inc., Waltham, MA, USA). The beads were washed twice with MagMAX cell-free DNA wash solution, and once with 80% ethanol. The washed beads were dried before elution in 64 µL of ribonuclease (RNase)-free water.

Quantification of RNA extracted from urine samples was not performed by spectrophotometry because the extraction process involves the addition of two synthetic RNA molecules to each extraction, thereby making spectrophotometric results inaccurate. These two RNA molecules are poly-A RNA, which acts as a carrier molecule to increase the efficiency of extraction for low RNA concentrations, and an internal control (IC) RNA, which is used to evaluate and control for extraction quality and RT-qPCR efficiency. Instead, quantification of RNA was performed using RT-qPCR, and extraction quality was determined using the spiked IC RNA.

### 2.4. Assay Controls

Each Cxbladder assay involved the detection of five target genes and the control gene, with the assay run in duplex (three duplex assays per Cxbladder assay). Each plate included the following controls with each duplex RT-qPCR: (1) no template control in triplicate for each duplex reaction (RNase-free water; a negative control for contamination); (2) three control samples containing either a high or low concentration of the five genes of interest, or no RNA for the genes of interest (HEC, LEC, and negative extraction control [NEC]; Appendix A); and (3) an individual sample quality control (QC), where each sample was spiked with IC RNA, a synthetic RNA transcript containing a unique sequence that lacks homology to known biological sequences prior to the extraction step control. If the RNA concentrations for the HEC or LEC samples fell outside the QC ranges (below or above the expected lower and upper limits, respectively), or the NEC sample showed contamination levels equal to the limit of quantification (LOQ), or the IC RNA concentrations were outside the expected range (log_10_1.55–2.90 copies/μL extracted RNA), then the PCR plate failed the QC check.

### 2.5. Reverse Transcription-Quantitative PCR

The Cxbladder assays used RT-qPCR to assess the mRNA expression of five biomarker genes (*CDK1*, *MDK*, *IGFBP5*, *HOXA13*, and *CXCR2*). Each assay was run as three duplex reactions (two genes per well: well 1 = *MDK* + *IGFBP5*; well 2 = *CDK1* + *HOXA13*; well 3 = *CXCR2* + IC RNA), with each duplex reaction performed in duplicate. Fluorescence color compensation was conducted using the LightCycler^TM^ 480-II (Roche, Basel, Switzerland) to account for reporter dye interference.

Individual biomarker raw data were initially defined as a quantitative cycle (Cq) value (the cycle of PCR where the fluorescence signal significantly exceeds background fluorescence noise). The standard curve of reference RNA was used to convert the Cq value for each biomarker to an mRNA concentration in copies/μL. The parameters for the standard curve of the reference RNA and the confirmation of the RNA concentrations of the known RT-qPCR controls (using the standard curve) were required to be within the defined quality control parameters (as shown in Appendix A) or the PCR plate failed QC.

### 2.6. Cxbladder Score Calculation

The Cxbladder Detect assay used a logistic regression classifier, based on the fluorescence data from the RT-qPCR analysis, to generate a score (scale 0.00–1.00) that indicates the probability of the patient having UC. The reportable range was determined during creation of the classifier scores, based on a training set population of 317 patients (274 were UC-free according to cystoscopy) [11]. Cxbladder Detect scores of ≥0 to <0.12 are categorized as normal gene expression (low risk of UC), scores of ≥0.12 to <0.23 as elevated gene expression (intermediate risk of UC), and scores of ≥0.23 to ≤1.0 as high gene expression (high risk of UC). Standard thresholds were applied to the classifier score to assist the clinician in making a decision about the likelihood of cancer based on the test population data.

The Cxbladder Triage algorithm included four clinical variables (age, sex, hematuria status, and smoking history; classified as the P-index) [12], as well as the five biomarker mRNA concentrations (G-index). The creation of the classifier score range (scale 0.0–10.0) was based on samples from 587 patients, as previously described [12]. The Cxbladder Triage score categorizes patients with hematuria as either “negative” (score < 4.0; low risk of UC) or “further investigation”/“not negative” (score ≥ 4.0; standard clinical workup is warranted).

The Cxbladder Monitor algorithm additionally included the status of the patient’s previous tumor (primary diagnosis or recurrent tumor) and the time since the previous tumor (in years) in addition to the five biomarker mRNA concentrations to provide the test score (scale 0.0–10.0). The Cxbladder Monitor score range was defined using a development set of 763 patients, as previously described [13]. Cxbladder Monitor provides a binary outcome for patients with previously diagnosed UC, whereby patients with a score of <3.5 are categorized as “rule out” (low risk of recurrent UC) and samples with a score of ≥3.5 are categorized as “further investigation” (high risk of recurrent UC).

### 2.7. Linearity

To assess the linearity of the individual biomarker mRNA concentrations used to generate the Cxbladder score, 13 RNA samples of different known concentrations that covered the expected range in samples (from beyond the LOQ to the limit of detection [LOD]) were analyzed to determine their measured concentrations using the reference RNA standard curve. Each standard curve was run in triplicate over two plates and the measured concentrations were plotted against the known concentrations.

Simple linear regression models were fitted for each gene. The null hypothesis was that the linear regression was valid for the whole diluted range. The null hypothesis was rejected if the R-squared value was <0.99. Linear and quadratic models were used to test for curvature in the reported range.

### 2.8. Limit of Quantification

Using the linearity assay above, LOQ was defined as 2 × the lowest linear concentration in attograms (ag; 10^−18^ g)/μL (expressed as RNA copies/mL urine, where one copy equated to 0.44456 ag) on the reference RNA standard curve for all five biomarkers (10.71 ag/μL, 24 copies/μL [log_10_ 1.385 copies/μL]), and was chosen to ensure near linearity of the standard curve.

### 2.9. Analytical Sensitivity

Analytical sensitivity (or LOD) was defined as the lowest RNA concentration that could be used to detect the biomarkers in 95% of assays performed. The numbers of total and amplified wells were counted for each gene. Assuming the number of wells had Poisson distribution (with rate = lambda [λ] × concentration), then the LOD was −log(0.05)/λ, and the maximum likelihood estimator for λ could be computed. The reference RNA was diluted to 3.2, 0.8, and 0.2 ag/µL (corresponding to 7.20, 1.80, and 0.45 RNA copies/mL, respectively). Reference RNA at 3.2 ag/μL was added to 24 wells, while RNA at 0.8 and 0.2 ag/µL was added to 28 wells each per duplex RT-qPCR, totaling 80 wells that spanned the LOD range. Based on the number of wells with “no signal”, the mRNA concentration could be computed, and the LOD corresponded to a 77.6% probability of detection in each of two replicate wells (i.e., [1 − 0.776]^2^ = 0.05).

### 2.10. Analytical Specificity

To determine the analytical specificity of the extracted RNA from biological interference, high-concentration samples were mixed individually with selected amounts of the following potentially interfering substances: red blood cells (RBC; 1 × 10^4^–1 × 10^9^ cells/mL); protein (0.01–10.00 mg/mL bovine serum albumin [BSA]); glucose (0.50 and 0.25 mg/mL); urea salts (75.0 and 37.5 mg/mL); bacteria (1000, 5000, and 10,000 cells/mL *Escherichia coli*); and yeast (10,000, 50,000, and 100,000 colony-forming units [cfu]/mL). Samples within the expected level of gene variance and where the Cxbladder score was within the 95% confidence interval (CI) for the control were considered acceptable.

To determine the analytical specificity of the RT-qPCR from process-related interference, the following potentially interfering substances were evaluated: absolute ethanol (0.230–3.750%); Cxbladder stabilizing reagent (0.012–3.000%); acetone (0.234–3.750%); MagMAX wash buffer (0.230–3.750%); and MagMAX bead solution (1.56–25.00%). For each potentially interfering substance, the starting concentration of inhibitor was combined, as a proportion of the final RT-qPCR volume, with reference RNA included at high and low concentrations.

### 2.11. Analytical Accuracy

The accuracy of the Cxbladder assays was determined by assessing the differences in the measured RNA concentrations in the controls versus the known concentrations using the extraction efficiency determined by the spiked extraction IC RNA. Data were collected from 20 RT-qPCRs run over ≥20 working days, consisting of 165 HEC and 65 LEC replicates.

To assess accuracy, a 1013 base pair (bp) reference RNA transcript, containing ~200 bp of each target sequence, was synthetically constructed and purified before its concentration was determined according to the Beer–Lambert law and spectrophotometry peak absorbance at 260 nm. From this concentration and the known nucleotide sequence, the actual transcript copy number was calculated utilizing the known molecular weight of each nucleotide. This copy number was used to determine the expected concentration, and the purified reference RNA was added to Cxbladder stabilizing reagent to create the HEC and LEC samples. A bias between the measured and known concentrations was expected, due to the RNA extraction process being unable to achieve 100% extraction efficiency. Acceptable inaccuracy was <15% for the linear region of the assay (>LOQ) and <20% for RNA concentrations that were <2 × LOQ.

The diagnostic sensitivity, specificity, positive predictive value (PPV), and NPV were calculated for Cxbladder Detect, Triage, and Monitor by comparison of clinical sample results against the cystoscopy-based diagnosis (see Appendix A for equations). For the accuracy validation analyses, 485 clinical samples were used for Cxbladder Detect (66 [13.6%] positive and 419 [86%] negative for UC by cystoscopy) [11], 587 samples for Triage (73 [12%] positive, 514 [88%] negative) [12], and 1036 samples for Monitor (156 [15%] positive, 880 [85%] negative) [13].

### 2.12. Analytical Precision

For the precision analysis, intra-assay and inter-assay variability of the genes in the Cxbladder Detect, Triage, and Monitor assays were assessed. For all genes, the acceptance criterion for precision was to have a total co-efficient of variation (CV%) of <10%.

Inter-assay variability was determined from 165 HECs and 65 LECs run over 20 plates. Intra-assay variability was determined by testing at least two HECs and at least two LECs run over ≥20 days by more than three operators. The data consisted of HEC samples from 14 plates of two replicates, >21 HEC samples over six plates, 19 plates containing two LEC samples, and one plate containing 23 LEC samples. Linear random effects models were fitted to estimate the intra-assay and inter-assay variability using R package LME4. Precision was calculated in the form of standard deviation (SD), and the bootstrapped SD was used to obtain the 95% CI of the estimation.

Lot-to-lot extraction reagent variation was evaluated by testing two sets of unqualified reagents against qualified reagents, using HEC and LEC samples in triplicate for each set. RT-qPCR reagent lot-to-lot variation was determined by testing two sets of unqualified versus existing qualified RT-qPCR reagents, using standard curves and RNA controls (1000, 120, and 30 ag RNA, giving Cxbladder Detect scores of 0.890, 0.332, and 0.055, respectively). Lot-to-lot variation was considered acceptable if the score variation was within the limits identified for an individual PCR plate.

The precision of the Cxbladder assays was further assessed by determining how well the assays performed in two laboratories (Pacific Edge Diagnostics, USA, Ltd. [PEDUSA], Hummelstown, PA, USA and Pacific Edge Diagnostics NZ [PEDNZ], Dunedin, NZ). Total variability was assessed at the gene concentration using HEC and LEC samples, and information was compared between laboratories (acceptable total CV% ≤ 10%).

### 2.13. Reproducibility

The inter-laboratory comparison between the NZ laboratory (PEDNZ) and the US laboratory (PEDUSA) was assessed for each Cxbladder assay. A random sample set from the PEDNZ validation was used to confirm the reproducibility of Cxbladder Detect (*n* = 153), Triage (*n* = 81), and Monitor (*n* = 41) at PEDUSA. Acceptable variability was defined as achieving ≥ 80% concordance for all clinical sample results.

### 2.14. Sample Stability

A mixed effects analysis of variance was used to assess trends over time for HEC and LEC RNA reference samples stored at 4 °C (2–8 °C), ambient room temperature (RT = 18–22 °C), or 30 °C. Statistical comparisons of sample scores were conducted using the *t*-test, where *p*-values of <0.05 were considered significant.

### 2.15. Statistical Analysis

Statistical analyses were conducted using R, version 4.3.1.

## 3. Results

### 3.1. Stability

In total, 62 samples each of HECs and LECs were added into Cxbladder stabilizing reagent and stored at 4 °C, RT, or 30 °C for 2–14 days. Day 14 samples were excluded due to standard curve efficiency failure (not instability), as Cxbladder scores were not calculable without a standard curve. Therefore, the difference in score between T0 and each timepoint was assessed over Days 2–12 at each temperature. The mixed effects analysis of variance found no evidence of a trend towards change in score over time at any of the temperatures assessed, with mean ± SD scores of 0.9352 ± 0.0181 for HEC and 0.0756 ± 0.0220 for LEC samples. The distribution of scores for HEC and LEC over time and across all temperatures were within the variation expected from random sampling. Therefore, samples collected and stored in Cxbladder stabilizing reagent were reliable and robust for a minimum of 12 days.

The freeze/thaw stability of the Cxbladder stabilizing reagent showed a maximum difference of 0.068 in Cxbladder Detect scores between the non-freeze/thaw reagent aliquots and those that had undergone five freeze/thaw cycles (Appendix A), which was considered acceptable. The change in the freeze/thaw reagent scores showed that Cxbladder stabilizing reagent remained stable for up to five freeze/thaw cycles.

The reference RNA showed a systematic shift in RT-qPCR Cq values for all five biomarker genes after three freeze/thaw cycles. This variation was <0.32 × Cq for *HOXA13* and <0.19 × Cq for *CDK1*, *MDK*, *IGFBP5*, *CXCR2*, and IC RNA (Appendix A). This variation was within the expected level of noise and linearity, and the Cxbladder score integrity was maintained. This shows that the reference RNA remained stable for up to three freeze/thaw cycles.

### 3.2. Linearity of RT-qPCR

The regression coefficients (*r*^2^) for the standard curves were >0.99 for each of the five individual biomarkers and the IC RNA (Figure 1). This verified the linearity of the RT-qPCR used in the Cxbladder assays throughout the tested range. The comparison between linear and quadratic models indicated that the relationship was linear at concentrations above the LOQ.

### 3.3. Analytical Sensitivity

Using Cxbladder Detect, the LOD was 12.5 RNA copies/mL urine for the *CDK1* and *MDK* biomarkers, 21.4 RNA copies/mL for *HOXA13*, 31.1 RNA copies/mL for *IGFBP5*, and 68.9 RNA copies/mL for *CXCR2* (Table 1).

### 3.4. Analytical Specificity

Analytical specificity was assessed using each individual gene and its inverse logistic space score from the Cxbladder algorithm. Extraction controls contaminated with bacteria (1000–10,000 cells/mL), glucose (0.25 or 0.50 mg/mL), urea salts (37.5 mg/mL), or yeast (10,000–100,000 cfu/mL) did not interfere with Cxbladder performance (Table 2). Protein was tolerated up to BSA 0.63 mg/mL in all extraction controls, while BSA 2.5 mg/mL inhibited performance in the LEC samples, but did not affect the HEC samples. BSA at concentrations of 10.0 mg/mL caused failure of IC RNA extraction in both the LEC and HEC samples, although the difference in overall Cxbladder score was not statistically significant. HEC samples contaminated with RBCs showed no interference up to 1 × 10^6^ cells/mL, but at higher concentrations, the Cxbladder scores showed a significant shift (in the negative direction). When reviewing the last 2 years of data from the PEDNZ and PEDUSA laboratories, the acceptable levels of inhibition that may come from these sources of contamination were 0.9% and 4.4%, respectively.

Of the process-related potentially interfering substances, Cxbladder solution was the most inhibitory reagent, requiring >0.750% of total RT-qPCR volume to interfere with the reaction for the LEC samples, followed by MagMAX wash buffer (>1.875%), absolute ethanol (>3.750%), acetone (>3.750%), and MagMAX beads (>25.000%; Appendix A). However, as Cxbladder stabilizing reagent is used early in the RNA extraction process, the actual risk of interference from Cxbladder stabilizing reagent in the downstream RT-qPCR is expected to be minimal. Any interference would be detected by the IC RNA included in every assay.

### 3.5. Analytical Accuracy

The results of the accuracy analysis for the measurement of RNA concentrations in the Cxbladder Detect assay are shown in Table 3. The loss of accuracy was ≤10.63% for the HEC samples, and ≤9.62% for LEC samples, and showed reliable RNA concentration measurements, withing the assay QC limits.

### 3.6. Analytical Precision

In the precision analysis, intra- and inter-assay variation at the individual biomarker gene level was highest in the LEC samples (low RNA concentrations; Table 4). The conservative 95% CIs included all sources of variation and ensured that the probability of misclassification due to uncontrolled variation across all biomarker genes was 4.384–8.011% for the LEC samples and 2.235–3.129% for HEC samples.

The precision of the Cxbladder assays was confirmed at the PEDUSA laboratory by calculating the total assay CV%. Cxbladder had an intra-assay variability of 1.04–6.85% and an inter-assay variability of 0.58–4.66% across the HEC and LEC samples and a total CV% of 1.47–8.29% (Appendix A).

### 3.7. Reproducibility

The inter-laboratory comparisons between PEDNZ and PEDUSA showed >85% concordance in clinical results for all three assays (86.27% for Cxbladder Detect, 92.59% for Triage, and 85.37% for Monitor).

## 4. Discussion

This analytical validation study demonstrated that the Cxbladder assays can accurately and reproducibly quantify the mRNA expression of five biomarker genes, with an analytical sensitivity or LOD of 12.5–31.1 RNA copies/mL urine for the four biomarkers correlated with UC (*CDK1*, *MDK*, *IGFBP5*, and *HOXA13*), and 68.9 copies/μL for the inflammatory biomarker *CXCR2*. All pre-specified analytical criteria, including linearity, analytical sensitivity, specificity, accuracy, and precision, were met.

The Cxbladder assays are multi-gene RT-qPCR assays that are designed to aid clinicians in the risk stratification of patients presenting with hematuria and those with previously treated UC who are undergoing surveillance. Cxbladder Detect is intended to provide moderate sensitivity and high specificity when screening for the presence or absence of UC in patients undergoing evaluation of hematuria, while Cxbladder Triage and Monitor are designed for high sensitivity and NPV to safely rule out UC in patients presenting with hematuria or a history of UC, respectively.

The accuracy of RNA concentration measurement showed an accuracy of >89% for all genes in both HEC and LEC samples (i.e., high and low RNA concentrations), with inaccuracy ranging from −2.23% to +10.63%.

The clinical accuracy of each Cxbladder assay was validated by comparison against the known cystoscopy-based diagnosis [11,12,13]. Cxbladder Detect scores for high (≥0.23) and elevated (≥0.12 to <0.23) gene expression samples had a sensitivity of 77% and 82%, respectively, as well as high specificity (94% and 83%) and NPVs (96% for both). Cxbladder Triage showed high sensitivity (95%) and NPV (98%), but lower specificity (46%). The sensitivity of Cxbladder Triage is higher than that of Detect due to the incorporation of four clinical risk factors into the algorithm that calculates the Triage test score [12]. These accuracy findings are consistent with those reported in a 2023 clinical validation study, in which the sensitivity, NPV, and specificity were 74%, 97%, and 82%, respectively, for Cxbladder Detect, and 89%, 99%, and 63%, respectively, for Cxbladder Triage [15]. In a 2021 systematic review of novel urinary biomarker tests, Cxbladder Detect and Triage had a pooled sensitivity of 97% for initial diagnosis of high-grade tumors [19]. In the current analysis, Cxbladder Monitor demonstrated high sensitivity (91%) and NPV (96%), albeit with low specificity (39%), similar to the internal validation of Monitor, in which the assay had a sensitivity, NPV, and specificity of 93%, 97%, and 34%, respectively [13]. The sensitivity and specificity of Cxbladder Monitor in the current study were also in line with the pooled sensitivity and NPV reported in the 2021 systematic review (91% and 98%, respectively) [19].

The analytical validity of the Cxbladder assays was confirmed at a second laboratory (PEDUSA), with maximum intra- and inter-assay variability of 4.66% and 6.85%, respectively, and a total maximum CV% of 8.29%. Concordance between the two laboratories (PEDNZ and PEDUSA) remained high (>85%) for all three assays based on the 95% CIs of the output results.

One of the limitations of this study was the use of RNA reference-spiked normal samples (rather than clinical samples) for the stability, linearity, analytical sensitivity, and specificity analyses; however, this limitation was mitigated by the use of blinded clinical samples for the validation of the clinical accuracy of each Cxbladder assay. Another limitation was that the Cxbladder assays did not include detection of other known genetic markers of UC, such as fibroblast growth factor receptor 3 (*FGFR3*) and telomerase reverse transcriptase (*TERT*) [20]. Ongoing studies have since led to the development of enhanced Cxbladder Detect and Triage assays [15], in which six DNA single nucleotide polymorphisms from *FGFR3* and *TERT* have been integrated to further improve risk stratification in patients with hematuria. Analytical and clinical validation of the enhanced Cxbladder Detect assay is currently ongoing, but a previous study found that this assay had a sensitivity, specificity, and NPV of 97%, 90%, and 99.7%, respectively [15]. Lastly, our study may have also been limited by its analytical validation design, meaning that it did not provide clinical validation or describe the clinical utility for these assays. However, previous studies have demonstrated the clinical validity of Cxbladder Detect, Triage, and Monitor [11,12,13], and the clinical utility of Cxbladder Triage and Monitor [21,22].

## 5. Conclusions

This analytical validation study demonstrated that the Cxbladder assays can accurately and reproducibly detect the mRNA expression of five biomarker genes that are associated with UC, with Cxbladder Detect having high sensitivity, specificity, and NPV, and the rule-out tests Cxbladder Triage and Monitor having higher sensitivity and NPV. These assays will aid clinicians in the risk stratification of patients presenting for evaluation of hematuria or those who are undergoing surveillance for recurrent UC, thereby enabling more accurate identification of patients at low or high risk of UC.

## Figures and Tables

**Figure 1 diagnostics-14-02061-f001:**
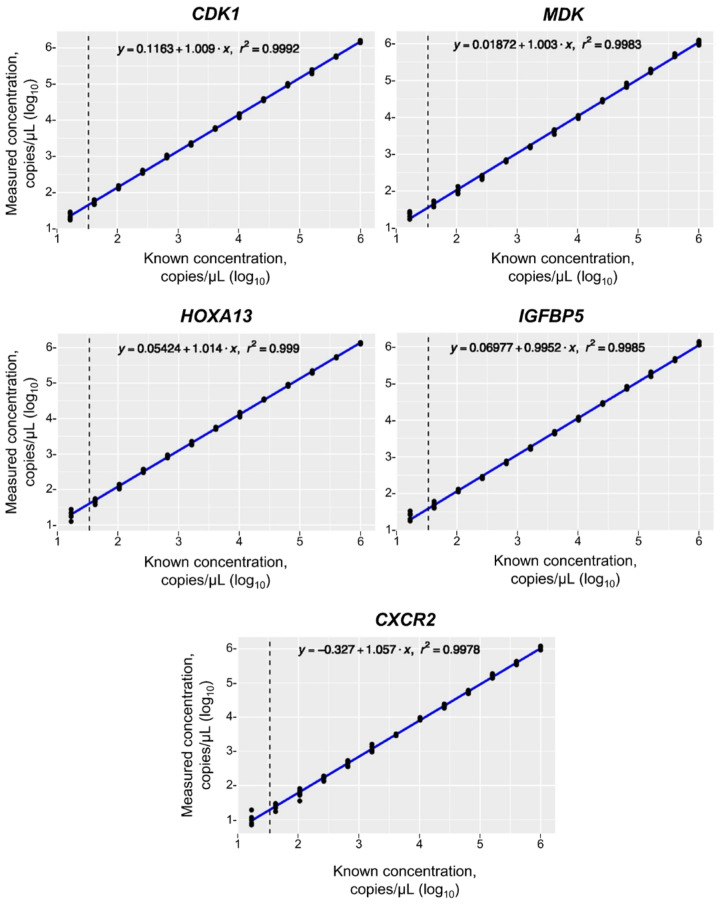
Measured versus known RNA concentrations for the five individual biomarker genes. The vertical dotted line in each curve represents the log10 LOQ for the five biomarkers (1.46 copies/μL). *CDK1*, cyclin-dependent kinase 1; *CXCR2*, C-X-C motif chemokine receptor 2; *HOXA13*, Homeobox A13; *IGFBP5*, insulin-like growth factor binding protein 5; LOQ, limit of quantification; *MDK*, midkine; *r*^2^, regression coefficient.

**Table 1 diagnostics-14-02061-t001:** Limit of detection (analytical sensitivity) for the five individual biomarker genes and internal control RNA in the Cxbladder assay.

Biomarker	Estimated LOD [95% CI]		
	RNA Mass (ag) per μL Urine	RNA Copies per mL Urine	Log_10_ RNA Copies per μL Urine
*CDK1*	0.496 [0.306, 0.712]	12.5 [7.7, 17.9]	1.116 [0.690, 1.600]
*MDK*	0.845 [0.535, 1.173]	12.5 [7.7, 17.9]	1.901 [1.200, 2.640]
*IGFBP5*	0.794 [0.496, 1.121]	31.1 [19.5, 44.0]	1.787 [1.120, 2.520]
*HOXA13*	0.764 [0.496, 1.075]	21.4 [13.9, 30.1]	1.719 [1.120, 2.420]
*CXCR2*	2.733 [1.899, 3.961]	68.9 [47.9, 99.9]	6.149 [4.270, 8.910]
IC RNA	1.642 [1.121, 2.344]	41.2 [28.1, 58.8]	3.695 [2.520, 5.270]

ag, attogram (10^−18^ g); CI, confidence interval; *CDK1*, cyclin-dependent kinase 1; *CXCR2*, C-X-C motif chemokine receptor 2; *HOXA13*, Homeobox A13; IC, internal control; *IGFBP5*, insulin-like growth factor binding protein 5; LOD, limit of detection; *MDK*, midkine.

**Table 2 diagnostics-14-02061-t002:** Analytical specificity of Cxbladder Detect with control samples when mixed with potentially interfering substances.

Contaminant	Cxbladder Score		*p*-Value ^b^
	Mean	Difference ^a^	
RBCs, cells/mL			
HEC sample	0.8250	–	–
1 × 10^4^	0.8400	+0.0150	0.2150
1 × 10^5^	0.8133	−0.0117	0.3933
1 × 10^6^	0.8300	+0.0050	0.7499
1 × 10^7^	0.7517	−0.0733	0.0003
1 × 10^8^	0.4133	−0.4117	0.0254
1 × 10^9^	0.0867	−0.7383	0.0000
LEC sample	0.0783	–	–
1 × 10^4^	0.0820	+0.0037	0.6924
1 × 10^5^	0.0783	0.0000	1.0000
1 × 10^6^	0.0983	+0.0200	0.0447
1 × 10^7^	0.0833	+0.0050	0.5626
1 × 10^8^	0.2333	+0.1550	0.2399
1 × 10^9^	0.4650	+0.3867	0.5583
Protein (BSA), mg/mL			
HEC sample	0.5850	–	–
0.01	0.5717	−0.0133	0.3677
0.04	0.5875	+0.0025	0.7718
0.16	0.5920	+0.0070	0.4146
0.63	0.5980	+0.0130	0.3308
2.5	0.6040	+0.0190	0.2593
10.0	0.6500	+0.0650	0.2486
LEC sample	0.1267	–	–
0.01	0.1175	−0.0092	0.1782
0.04	0.1083	−0.0182	0.0739
0.16	0.1260	−0.0007	0.9423
0.63	1.1460	+0.0193	0.0616
2.5	0.2580	+0.1313	0.0048
10.0	0.2500	+0.1233	0.3660
Glucose, mg/mL			
Control sample	0.8225	–	–
0.25	0.7750	−0.0475	0.0128
0.50	0.7500	−0.0725	0.2911
Salts (urea), mg/mL			
Control sample	0.8225	–	–
37.5	0.7675	−0.0550	0.0946
75.0	0.7980	−0.0245	0.0349
Bacteria (*E. coli*), cells/mL			
Control sample	0.8675	–	–
1000	0.8725	+0.0050	0.6554
5000	0.8740	+0.0065	0.4517
10,000	0.8750	+0.0075	0.3884
Yeast (cfu/mL)			
Control sample	0.8675	–	–
10,000	0.8650	−0.0025	0.8238
50,000	0.8800	+0.0125	0.1411
100,000	0.8500	−0.0175	0.0867

^a^ Calculated by subtracting the mean score for the control sample from that of each contaminated sample. ^b^ Two-sided *t* test. BSA, bovine serum albumin; cfu, colony-forming units; *E. coli*, *Escherichia coli*; HEC, high-extraction control; LEC, low-extraction control; RBC, red blood cell.

**Table 3 diagnostics-14-02061-t003:** Accuracy of RNA concentrations across the five individual biomarkers and internal control RNA in the Cxbladder Detect assay, as measured in 165 HEC and 63 LEC across 20 extractions and RT-qPCR plates.

Biomarker Gene	Log_10_ RNA Concentration (Copies/μL)			Inaccuracy (%) ^a^
	Measured (SD)	Known	Difference	
*CDK1*				
HEC	3.292 (0.074)	3.47	+0.178	+5.13
LEC	2.147 (0.094)	2.10	−0.047	−2.24
*MDK*				
HEC	3.106 (0.097)	3.47	+0.364	+10.49
LEC	2.018 (0.130)	2.10	+0.082	+3.90
*IGFBP5*				
HEC	3.101 (0.092)	3.47	+0.369	+10.63
LEC	2.006 (0.106)	2.10	+0.094	+4.48
*HOXA13*				
HEC	3.215 (0.084)	3.47	+0.255	+7.35
LEC	2.037 (0.106)	2.10	+0.063	+3.00
*CXCR2*				
HEC	3.106 (0.071)	3.47	+0.364	+10.49
LEC	1.898 (0.152)	2.10	+0.202	+9.62

^a^ Calculated using the following equation: (known concentration—measured concentration) ÷ known concentration × 100. *CDK1*, cyclin-dependent kinase 1; *CXCR2*, C-X-C motif chemokine receptor 2; *HOXA13*, Homeobox A13; HEC, high-extraction control; *IGFBP5*, insulin-like growth factor binding protein 5; LEC, low-extraction control; *MDK*, midkine; RT-qPCR, reverse transcription-quantitative polymerase chain reaction; SD, standard deviation.

**Table 4 diagnostics-14-02061-t004:** Intra- and inter-assay variability across 20 RT-qPCR runs for high- and low-extraction controls for the Cxbladder assays.

	Variance ± SD (CV%)		
	Intra-Assay Variability	Inter-Assay Variability	Total Variability
HEC (logit score)	0.000 ± 0.021	0.001 ± 0.035	0.002 ± 0.040
* CDK1*	0.001 ± 0.027 (0.832)	0.005 ± 0.068 (2.075)	0.005 ± 0.074 (2.235)
* MDK*	0.002 ± 0.045 (1.437)	0.007 ± 0.086 (2.779)	0.009 ± 0.097 (3.129)
* IGFBP5*	0.001 ± 0.036 (1.176)	0.007 ± 0.085 (2.736)	0.009 ± 0.092 (2.978)
* HOXA13*	0.001 ± 0.036 (1.107)	0.006 ± 0.077 (2.383)	0.007 ± 0.084 (2.627)
* CXCR2*	0.002 ± 0.071 (1.458)	0.003 ± 0.054 (1.745)	0.005 ± 0.071 (2.274)
IC RNA	0.001 ± 0.038 (1.244)	0.012 ± 0.112 (3.664)	0.014 ± 0.118 (3.869)
LEC (logit score)	0.001 ± 0.025	0.002 ± 0.033	0.002 ± 0.047
* CDK1*	0.003 ± 0.057 (2.660)	0.006 ± 0.075 (3.485)	0.009 ± 0.094 (4.384)
* MDK*	0.006 ± 0.075 (3.696)	0.011 ± 0.106 (5.255)	0.017 ± 0.130 (6.424)
* IGFBP5*	0.004 ± 0.061 (3.017)	0.007 ± 0.087 (4.316)	0.011 ± 0.106 (5.266)
* HOXA13*	0.005 ± 0.067 (3.312)	0.007 ± 0.082 (4.018)	0.011 ± 0.106 (5.207)
* CXCR2*	0.006 ± 0.081 (4.246)	0.017 ± 0.129 (6.794)	0.023 ± 0.152 (8.011)
IC RNA	0.004 ± 0.066 (2.327)	0.005 ± 0.074 (2.596)	0.010 ± 0.100 (3.486)

*CDK1*, cyclin-dependent kinase 1; CV%, coefficient of variation; *CXCR2*, C-X-C motif chemokine receptor 2; HEC, high-extraction control; *HOXA13*, Homeobox A13; IC, internal control; *IGFBP5*, insulin-like growth factor binding protein 5; LEC, low-extraction control; logit, inverse logistic space; *MDK*, midkine; RT-qPCR, reverse transcription-quantitative polymerase chain reaction; SD, standard deviation.

## Data Availability

The raw data supporting the conclusions of this article will be made available by the authors on upon reasonable request.

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
