# Peer review of "Analytical Validation of Cxbladder® Detect, Triage, and Monitor: Assays for Detection and Management of Urothelial Carcinoma"

_diagnostics, 2024, doi:10.3390/diagnostics14182061_

Round 1

Reviewer 1 Report

Comments and Suggestions for Authors

The authors studied the clinical utility of Cxbladder® Detect, as a tool to detect UC and validated their findings in an external cohort. This is a well-conducted work and has important clinical implications if replicated in a broader and more diverse population. A few minor comments are as follows: 

1- why did the authors choose these specific genes for their panel? why not including other genes like FGFR3 and TERT which were previously studied for the same purpose? Do the authors think it would be important to compare  Cxbladder Detect with previously reported assays? A comment answering each of these questions should be specifically added to the results or/and discussion sections of the manuscript.

2-what were the study's limitations?

3- what is the authors'suggestions for clinical use of Cxbladder detect with respect to previously treated non-muscle invasive patients? HOw soon after intravesical instilations and how often?

4-Overall, do the authors think that Cxbladder detect is a more robust non-invasive method for detecting UC compared to urine cytology?

Reviewer 2 Report

Comments and Suggestions for Authors

The authors conducted a validation of the CxBladder assay. Their pre-specified acceptance criteria included the fundamental aspects of the assays (sample and reagent stability, RNA extraction quality, RT-qPCR linearity, and analytical sensitivity and specificity), accuracy and precision, and reproducibility between laboratories. They looked at potentially interfering substances such as red blood cells, protein, urea salts, glucose, and bacteria, as well as other reagents such as ethanol, stabilizing reagent, acetone, MagMAX wash buffer, and MagMAX bead solution.

The methodology was made very explicit and detailed, even at the molecular level. They had a robust study with almost 900 samples. Stability was assessed and consistent for a minimum of 12 days.  A total of 5 freeze-thaw cycles resulted in stability.  The contaminants did not impact CxBladder performance.

They also incorporated secondary laboratories (PEDUSA) to explore analytical validity.

This is a very well-conducted analysis utilizing sophisticated molecular biology techniques.

It would be insightful for the authors to include the CxBladder company standards for what is considered an acceptable sample, or at least to elucidate protocolized reasons for “rejecting” a sample, i.e. if too much protein or glucose or bacteria in the sample. Does this protocol concur with the findings of the present study about the acceptable levels of “contamination?”
